# Hotspot areas of risky sexual behaviour and associated factors in Ethiopia: Further spatial and mixed effect analysis of Ethiopian demographic health survey

Denekew Tenaw Anley[1]*, Melkamu Aderajew Zemene[1], Asaye Alamneh Gebeyehu[1], Natnael Atnafu Gebeyehu[2], Getachew Asmare Adella[3], Gizachew Ambaw Kassie[4], Misganaw Asmamaw Mengstie[5], Mohammed Abdu Seid[6], Endeshaw Chekol Abebe[5], Molalegn Mesele Gesese[2], Yenealem Solomon[7], Natnael Moges[8], Berihun Bantie[9], Sefineh Fenta Feleke[10], Tadesse Asmamaw Dejenie[11], Ermias Sisay Chanie[8], Wubet Alebachew Bayih[12,13], Natnael Amare Tesfa[14], Wubet Taklual[1], Dessalegn Tesfa[1], Rahel Mulatie Anteneh[1], Anteneh Mengist Dessie[1]

1 Department of Public Health, College of Health Sciences, Debre Tabor University, Debre Tabor, Ethiopia, 2 Department of Midwifery, College of Medicine and Health Science, Wolaita Sodo University, Wolaita Sodo, Ethiopia, 3 Department of Reproductive Health and Nutrition, School of Public Health, Woliata Sodo University, Woliata Sodo, Ethiopia, 4 Department of Epidemiology and Biostatistics, School of Public Health, Woliata Sodo University, Woliata Sodo, Ethiopia, 5 Department of Biochemistry, College of Health Sciences, Debre Tabor University, Debre Tabor, Ethiopia, 6 Department of Biomedical Science, Unit of Physiology, College of Health Science, Debre Tabor University, Debre Tabor, Ethiopia, 7 Department of Medical Laboratory Science, College of Health Sciences, Debre Tabor University, Debre Tabor, Ethiopia, 8 Department of Pediatrics and Child Health Nursing, College of Health sciences, Debre Tabor University, Debre Tabor, Ethiopia, 9 Department of Comprehensive Nursing, College of Health Sciences, Debre Tabor University, Debre Tabor, Ethiopia, 10 Department of Public Health, College of Health Sciences, Woldia University, Woldia, Ethiopia, 11 Department of Medical Biochemistry, College of Medicine and Health Sciences, University of Gondar, Gondar, Ethiopia, 12 Department of Maternal and Neonatal Health Nursing, College of Health Sciences, Debre Tabor University, Debre Tabor, Ethiopia, 13 Faculty of Medicine, Department of Epidemiology and Preventive Medicine, School of Public Health and Preventive Medicine, Nursing and Health Sciences, Monash University, Victoria, Australia, 14 School of Medicine, College of Health Science, Woldia University, Woldia, Ethiopia

* denekewtenaw7@gmail.com

**Data Availability Statement:** All relevant data are within the paper and its Supporting Information files.

## Abstract

### Introduction

Sexual behaviour needs to take a central position in the heart of public health policy makers and researchers. This is important in view of its association with Sexually Transmitted Infections (STIs), including HIV. Though the prevalence of HIV/AIDS is declining in Ethiopia, the country is still one of the hardest hit in the continent of Africa. Hence, this study was aimed at identifying hot spot areas and associated factors of risky sexual behavior (RSB). This would be vital for more targeted interventions which can produce a sexually healthy community in Ethiopia.

### Methods

In this study, a cross-sectional survey study design was employed. A further analysis of the 2016 Ethiopia Demographic and Health Survey data was done on a total weighted sample

**Funding:** The author(s) received no specific funding for this work.

**Competing interests:** The authors have declared that no competing interests exist.

of 10,518 women and men age 15–49 years. ArcGIS version 10.7 and Kuldorff's SaTScan version 9.6 software were used for spatial analysis. Global Moran's I statistic was employed to test the spatial autocorrelation, and Getis-Ord Gi* as well as Bernoulli-based purely spatial scan statistics were used to detect significant spatial clusters of RSB. Mixed effect multivariable logistic regression model was fitted to identify predictors and variables with a p-value ≤0.05 were considered as statistically significant.

## Result

The study subjects who had RSB were found to account about 10.2% (95% CI: 9.64%, 10.81%) of the population, and spatial clustering of RSB was observed (Moran's I = 0.82, p-value = 0.001). Significant hot spot areas of RSB were observed in Gambela, Addis Ababa and Dire Dawa. The primary and secondary SaTScan clusters were detected in Addis Ababa (RR = 3.26, LLR = 111.59, P<0.01), and almost the entire Gambela (RR = 2.95, LLR = 56.45, P<0.01) respectively. Age, literacy level, smoking status, ever heard of HIV/AIDS, residence and region were found to be significant predictors of RSB.

## Conclusion

In this study, spatial clustering of risky sexual behaviour was observed in Ethiopia, and hot spot clusters were detected in Addis Ababa, Dire Dawa and Gambela regions. Therefore, interventions which can mitigate RSB should be designed and implemented in the identified hot spot areas of Ethiopia. Interventions targeting the identified factors could be helpful in controlling the problem.

## Introduction

High-risk behaviors are defined as acts that increase the risk of disease or injury, which can subsequently lead to disability, death, or social problems [1]. Sexual behavior encompasses all activities that satisfy an individual's sexual needs. Sexual practices, sexual relationships, reproductive health, sexually transmitted infections (STIs), and contraception have all been investigated under sexual behavior [2].

Risky sexual behavior, also known as high-risk sexual behavior, is a sexual behavior that increases an individual's susceptibility to sexually transmitted diseases (STIs) like the human immunodeficiency virus (HIV), chlamydia, gonorrhea, syphilis, and trichomonas' disease [3–5]. Infections that go untreated can cause pelvic inflammatory disease, infertility, ectopic pregnancy, and chronic pelvic pain [6]. They can also lead to unfavorable pregnancy outcomes such as spontaneous abortion, stillbirth, premature birth, different congenital infections, and psychological discomfort [7]. Early sexual initiation, unprotected sexual intercourse, and multiple sexual partners (having sex with commercial sex workers) are all risky sexual behaviors. The intensity of such involvement ranges from non-sexual partnerships to unprotected sexual intercourse with several partners [8,9].

According to WHO, 333 million new cases of STIs occur worldwide each year, and at least 111 million of these cases occur in people under age 25 years. In developing countries, nearly half of all HIV infections occur in men and women younger than 25 years this data indicate that up to 60% of all new HIV infections are among 15 to 24 years [10].

In Ethiopia, the sexual and reproductive health of young people has become a major public health concern due to a high prevalence of sexually transmitted infections (STIs) like HIV/

AIDS among young people. It is estimated that young people aged 10–24 years constitute more than a third of the population, 26.5 million (33%) [11]. Surveillance data indicate that almost 50 percent of all new sexually transmitted infections are found in adolescents and young adults [12], Adolescents life are at risk because they do not have the information, skills, health services and support they need to go through sexual development during adolescent. Ethiopia is one of the developing African countries where HIV/AIDS is fueling and striking its population of all age including adolescents [3].

According to different literature documentation factors associated with risky sexual behavior include sex, age, educational status, occupational status, divorced parents, substance abuse disorders, average daily income, permissive attitudes, mood disorders, peer pressure, unpleasant childhood events like sexual abuse, exposure to sexual content of media, sexual trafficking, or maltreatment, previous residence, type of sleeping place, cigarette smoking, and benzene sniffing [3,13,14].

Even though the prevalence and associated factors were researched in many parts of Ethiopia [15–26], according to our literature search, there is minimal evidence in distinguishing those hot spot and cold spot areas in different parts of Ethiopia. Spatial analysis is crucial in Ethiopia, particularly for identifying hot spot areas, as it offers a comprehensive understanding of geographic patterns and relationships within datasets. Moreover, spatial analysis helps bridge the gap between data and actionable insights by visualizing complex spatial relationships, allowing for targeted interventions and resource allocation regarding to tackling RSB. In essence, spatial analysis offers a more holistic and precise understanding of spatial phenomena, making it an indispensable tool for evidence-based decision-making and addressing the unique challenges of RSB faced by Ethiopia. Hence, identifying hotspot regions and factors linked with those are explored here to avert difficulties in Ethiopia's reproductive age group.

Significant hurdles exist to effectively preventing and managing sexually transmitted diseases (STIs), including HIV. Knowing the hotspot areas and factors responsible for the problem helps prevent and control those high-risk behaviors and their consequences. Effective prevention requires not only knowing who is at risk but also understanding why they engage in risk behaviors, motivating them to reduce their risk, developing their knowledge and skills, improving their access to prevention in ways that are appropriate to them, and creating a supportive social and policy environment for behavioral change. As a result, the purpose of this study was to determine the hotspot areas of high-risk sexual behavior and associated factors in Ethiopia.

## Materials and methods

### Study design, setting and period

A cross-sectional survey study design was conducted in Ethiopia using 2016 EDHS. Ethiopia is located in the Horn of Africa and has 9 Regions and two administrative cities. The country lies completely within the tropical latitudes and is relatively compact, with similar north-south and east-west dimensions. The capital is Addis Ababa ("New Flower"), located almost at the centre of the country. Ethiopia's population growth rate is well above the global average and is among the highest in Africa. Birth and death rates for the country are also well above those for the world

### Source and study population

The source population was all women and men in the age of 15–49 years in Ethiopia, and the study population was those who had sexual contact 12 months preceding the survey.

### Eligibility criteria

In this study, people in the age of 15–49 years were included. However, people who didn't have sexual contact 12 months preceding the survey were excluded from the study.

### Sample size and sampling procedure

Women and men of 15-49years old who had sexual contact 12 months preceding the survey are study subjects. Weighted values were used to restore the representativeness of the sample data. Sample weights were calculated in individual records (IR) EDHS datasets. Final sample size of the study subjects was 10,518. The survey covered all nine regions and the two city administrations of Ethiopia. Participants were selected based on a stratified two-stage cluster sampling technique. The detailed sampling procedure was available in EDHS report from Measure DHS website (www.dhsprogram.com).

### Variables

**Outcome variable.** The outcome variable of this study is Risky Sexual Behaviour (RSB) (Yes/No). Individuals having multiple sexual partners and/or high-risk sexual partners, and/or no condom use in the past 12 months were classified as having RSB.

**Independent variables.** In this study, both individual and community level independent variables were considered. The individual-level factors included were; Age, literacy level, ever tested for HIV, Ever hear about HIV, marital status, wealth, age at first sex, smoking status and media exposure. Four variables, place of residence, region, community level literacy and community level poverty were considered as community level factors.

### Data collection tools and procedures

Rigorous data collection tools and procedures were used to gather comprehensive and reliable information on various health and demographic indicators of RSB. Through household surveys, structured questionnaires, and standardized measurements, a representative sampling and high data quality were ensured. Trained field personnel conduct face-to-face interviews, employing systematic sampling techniques to reach diverse populations across Ethiopia's regions and communities. The detailed data collection procedure was available in EDHS report from Measure DHS website (www.dhsprogram.com).

### Data management and analysis

Data were cleaned by STATA version 14.1 software and Microsoft excel. Before performing spatial analysis, the weighted proportion (using sample weight) of RSB and candidate explanatory variables data was exported to ArcGIS version 10.7.

### Spatial analysis

**Spatial autocorrelation.** We used Arc GIS 10.7 software for spatial autocorrelation and detection of hot spot areas. Spatial autocorrelation (Global Moran's I) statistic measure was used to assess whether RSB was dispersed, clustered, or randomly distributed in Ethiopia. Moran's I values close to − 1 indicates dispersed RSB, close to + 1 indicates clustered, and if Moran's I value zero indicates randomly distributed [27].

**Hot spot analysis.** The proportion of RSB in each cluster was taken as an input for hot spot analysis. Hot Spot Analysis (Getis-Ord Gi* statistic) of the z-scores and significant p-values tell the features with either hot spot or cold spot values for the clusters spatially. The hot

spot areas indicated that there was a high proportion of RSB, and the cold spot ones indicated that there was a low proportion.

**Spatial interpolation.** The spatial interpolation technique was used to predict RSB for unsampled areas based on sampled clusters. For the prediction of unsampled clusters, we used geostatistical ordinary Kriging spatial interpolation technique using ArcGIS 10.7 software.

**Spatial scan statistics.** Bernoulli based model spatial scan statistics was employed to determine the geographical locations of statistically significant clusters for RSB using Kuldorff's SaTScan version 9.6 software [28]. The scanning window that moves across the study area in which study subjects had RSB were taken as cases and those who had no RSB taken as controls to fit the Bernoulli model. We used Bernoulli model because the outcome variable was binary. The default maximum spatial cluster size of < 50% of the population was used as an upper limit, allowing both small and large clusters to be detected, and ignored clusters that contained more than the maximum limit with the circular shape of the window. Most likely clusters were identified using p-values and likelihood ratio tests based on 999 Monte Carlo replications.

## Model building

Due to the hierarchical nature of the 2016 EDHS data, where individuals are nested within the community, the assumptions such as independent of observations and equality of variance have been violated. Therefore, multilevel binary logistic regression was fitted for the study of determinants of RSB among the study subjects. Four models were used in the multilevel analysis. These models were fitted using a STATA command **xtmelogit**. The first model contained only the outcome variable which was used to check the proportion of RSB variability in the community. The second models contain only individual-level variables and the third model contains only community-level variables, whereas, in the fourth model, both the individual and community-level variables were adjusted simultaneously with the outcome variable. For model comparison, we used the Log-Likelihood Ratio (LLR). The model with highest log-likelihood was considered as the best-fitted model.

## Parameter estimation methods

**The fixed effects (a measure of association)** In the multilevel multivariable logistic regression model, fixed effect estimates measure the association between the odds of RSB of individual and community level factors with a 95% confidence interval. Univariable analysis was carried out, and a variable with P-value less than or equal to 0.25 was taken as the candidate variable for the multivariable analysis. Adjusted Odds Ratio (AOR) with 95%.

Confidence interval (CI) and P-value less than 0.05 were reported as a significant factor that affects RSB. Multicollinearity was checked using the Variance Inflation Factor (VIF), and VIF less than 10% was taken as no multicollinearity.

**Random-effects (measure of variation)** were estimated by median odds ratio (MOR), intra correlation coefficient (ICC), and Proportional Change in Variance (PCV). The diviance, MOR, ICC and and AIC of the models were estimated in stata softwere using the "**xtmrho**" package. Formulas described bellow can also be used to calculate the above mentioned parameteres.

The MOR is defined as the median value of the odds ratio between the area at the highest risk and the area at the lowest risk when randomly picking out two clusters.

MOR = exp.$[\sqrt{(2 \times VA)} \times 0.6745]$, or MOR $= e^{0.95\sqrt{VA}}$ where; VA is the area level variance [29–31].

The PCV reveal the variation in RSB among the study subjects explained by factors. The PCV is calculated as; $PCV = \frac{Vnull-VA}{V\,null} *100\%$ where; Vnull = variance of the initial model, and

VA = variance of the model with more terms. The ICC which reveals the variation of RSB between clusters is calculated as;

$ICC = \frac{VA}{VA+3.29} *100\%$, where; VA = area/cluster level variance.

### Ethical approval and consent to participate

The data used for the analysis and the authorized letter were obtained from the measure DHS program website (www.dhsprogram.com) after proper registration and request. The data was accessed on Jan 30, 2023. Besides, authors had no access to information that could identify individual participants during or after data collection.

## Results

### Background characteristics of the study subjects

A total of 10, 518 study subjects were included in this study. The mean age was found to be 30 years with standard deviation (SD) of 0.08. Most of them (9, 647 (92%)) were found to be married. The majority of them were having rural residence (8,656 (82%)). Most of the study subjects started sexual intercourse before 18 years of age, and half of them have never been tested for HIV. The unit of analysis for the community level factors were clusters. Among the 645 clusters, 643 were found to be eligible for this study in which men and women of 15–49 years had sexual contact 12 months preceding the survey. The community level poverty was found to be high (96.4%) (**Table 1**).

### Prevalence of RSB in Ethiopia, EDHS 2016

The overall prevalence of RSB in Ethiopia was estimated to be 10.21% with 95% CI of (9.64, 10.82) (**Fig 1**). The urban and rural prevalence of RSB was found to be about 22% and 6.2% respectively.

### Spatial autocorrelation

The spatial distribution of RSB was non-random in Ethiopia. This was evidenced by the Global Moran's I value of 0.82 (P-value <0.001) (**Fig 2**).

### Spatial distribution of RSB in Ethiopia

The spatial distribution of RSB was non-random in Ethiopia. This was evidenced by the Global Moran's I value of 0.82 (P-value <0.001). Each point on the map represents one census enumeration area which encompasses several cases of RSB. The red color indicates areas with a high prevalence of RSB, and the green color indicates areas with a low prevalence of RSB (**Fig 3**).

### Hot spot analysis

Hot spot areas of RSB were detected in the Gambela region, Addis Ababa, and western Dire Dawa (**Fig 4**).

### Spatial scan statistics analysis

In the spatial scan statistics analysis, 85 most likely clusters were detected spatially. The most likely clusters of RSB were detected in the western part of Addis Ababa, western part of Dire Dawa, and Gambela region of Ethiopia. From the most likely clusters, 53 of them were primary clusters located at 8.972907 N and 38.745529 E with 13.24 km radius. Individuals who live in

**Table 1. Sociodemographic characteristics of respondents (n = 10,518).**

| Characteristics | Category | Weighted frequency | Percentage |
|---|---|---|---|
| Age (years) | 15–19 | 724 | 6.88 |
| | 20–24 | 1,817 | 17.27 |
| | 25–29 | 2,462 | 23.41 |
| | 30–34 | 2,031 | 19.31 |
| | 34–39 | 1,639 | 15.58 |
| | 40–44 | 1,075 | 10.22 |
| | 44–49 | 770 | 7.32 |
| Education | Literate | 4,235 | 40.26 |
| | No formal education | 6,283 | 59.74 |
| Current working status | Working | 3,388 | 32.21 |
| | Not working | 7,130 | 67.79 |
| Religion | Orthodox | 4,422 | 42.04 |
| | Catholic | 80 | 0.76 |
| | Protestant | 2,306 | 21.92 |
| | Muslim | 3,538 | 33.64 |
| | Traditional | 101 | 0.96 |
| | Other | 72 | 0.68 |
| Marital status | Never married | 282 | 2.68 |
| | Married | 9,647 | 91.72 |
| | Widowed/separated | 588 | 5.59 |
| Wealth | Poor | 4,045 | 38.46 |
| | Middle | 2,118 | 20.13 |
| | Rich | 4,356 | 41.41 |
| Age at first sex | <18 years | 8,027 | 76.31 |
| | > = 18 years | 2,491 | 23.69 |
| Smoking status | Smoker | 85 | 0.81 |
| | Not smoker | 10,433 | 99.19 |
| Ever been tested for HIV | Yes | 5,155 | 49.01 |
| | No | 5,363 | 50.99 |
| Media exposure | Exposed | 234 | 2.22 |
| | Not exposed | 10,284 | 97.78 |
| Ever Heard of HIV/AIDS | Yes | 9,751 | 92.71 |
| | No | 766 | 7.29 |
| Community literacy | Low | 304 | 2.89 |
| | High | 10,213 | 97.11 |
| Community poverty | Low | 383 | 3.64 |
| | High | 10,135 | 96.36 |
| Residence | Urban | 1,862 | 17.71 |
| | Rural | 8,656 | 82.29 |
| Region | Tigray | 726 | 6.90 |
| | Afar | 96 | 0.91 |
| | Amhara | 2,557 | 24.31 |
| | Oromia | 4,042 | 38.43 |
| | Somali | 324 | 3.08 |
| | Benshangul | 116 | 1.10 |
| | SNNPR | 2,123 | 20.18 |
| | Gambela | 31 | 0.29 |
| | Harari | 25 | 0.24 |
| | Addis Ababa | 425 | 4.04 |
| | Dire Dawa | 53 | 0.51 |

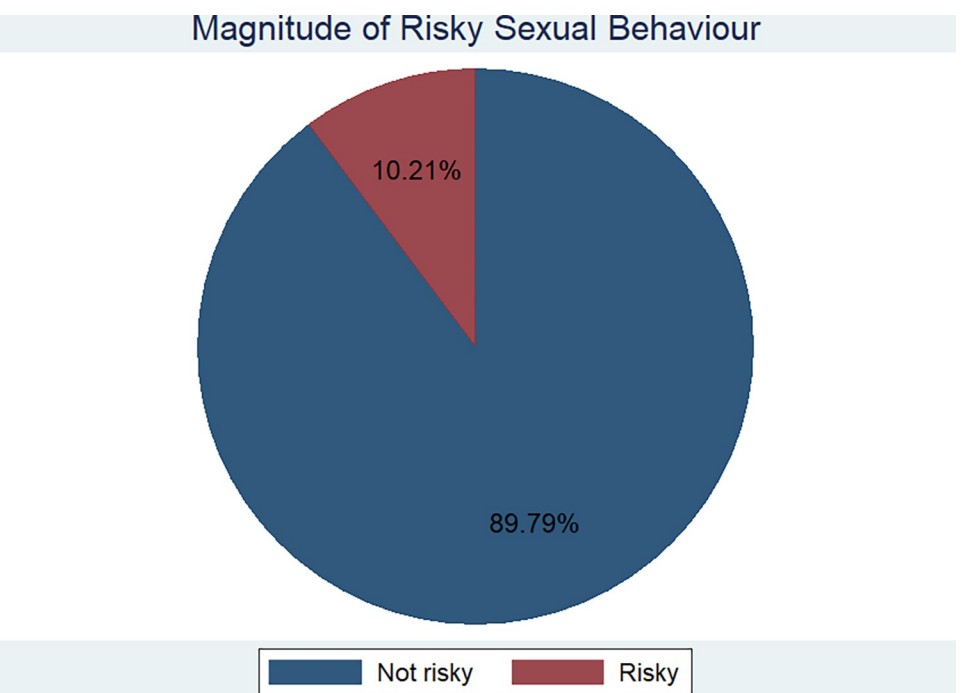

**Fig 1. Over all prevalence of risky sexual behaviour in Ethiopia, 2016.**

the primary clusters were 3.26 times more likely to have RSB as compared to outside the window (RR = 3.26, LLR = 111.59, P-value < 0.001) (**Fig 5**).

## Kriging interpolation of RSB

Based on EDHS 2016 sampled data, the Kriging interpolation predict the highest prevalence of RSB in some parts of Dire Dawa, Addis Ababa and Gambela (prevalence of RSB: 29.47–35.17%). Most parts of Amhara, Somalia and Afar regions were predicted to have the prevalence of RSB 6.6% 12.3%. Besides, the prevalence of RSB in most part of Tigray region was predicted to be 12.3%-18% (**Fig 6**).

## Factors associated with risky sexual behaviour

### Random effect analysis

In the null model, the ICC value was 25.3% (95% CI: 20.99%, 27.88%), indicated that about 25.3% of the overall variability in RSB was explained by the between cluster variation while the remaining 74.7% was attributed to the individual level variation. Besides, the Likelihood Ratio (LR) test was (LR test vs. logistic model: $X2(01) = 336.06$, p< 0.0001), which showed that the mixed-effect models were the best-fitted model for this data compared to the standard model.

The MOR for RSB was 2.74 in the empty model (Model 1); this showed that there was variation between communities (clustering) since MOR was 2.74 times higher than the reference (MOR = 1). The unexplained community variation in RSB decreased to MOR of 1.65 when all factors were added to the null model (empty model). As evidenced by the highest LLR, the lowest ICC and deviance the final model (mixed effect model) was found to be the best fit model explaining RSB in Ethiopia. Besides, the significant (p-value<0.001) likelihood ratio (lr) test of the four models verified improvement and the nested nature of the four models as we go from the null model (Model 1) to the full model (model 4) (**Table 2**).

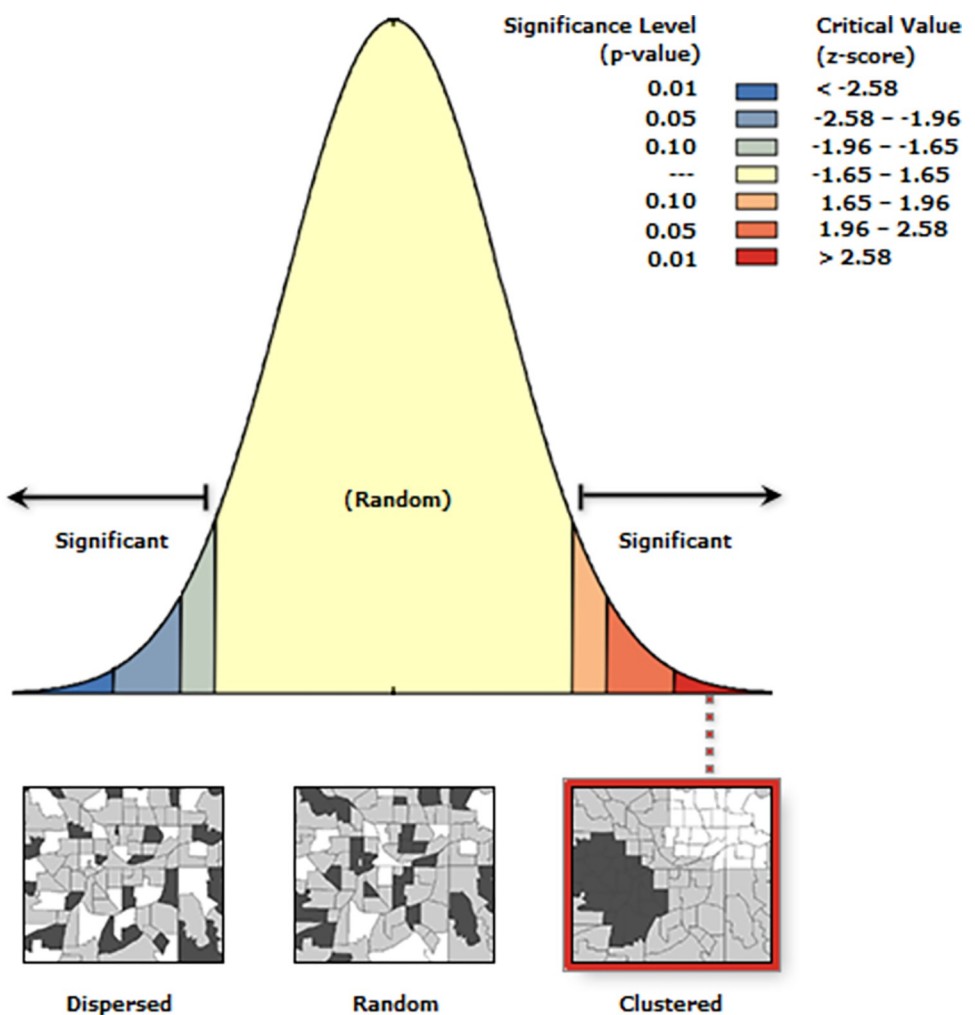

**Fig 2. Spatial clustering of risky sexual behaviour in Ethiopia, 2016.**

## Fixed effect analysis results

In the multivariable mixed-effect logistic regression; age, literacy level, smoking status, media exposure and ever heard of HIV/AIDS were found to be individual level associated factors of RSB in Ethiopia. Keeping other individual and community level factors constant, as age increases, the odds of having RSB decreases by about 2% (AOR: 0.978, 95% CI (0.961 0.995)). Individuals with no formal education were about 30% (AOR: **1.3; 95% CI: 1.02 1.42**) high likely to have RSB as compared to their counter parts. Besides, smokers were about 3.3 times more likely to have RSB as compared to the non-smokers (AOR: **3.28; 95% CI: 1.247, 8.631).** The odds of having RSB among individuals with no media exposure was about 2 times higher as compared to their counterparts (AOR: 1.908; 95% CI: 1.204, 3.024). Furthermore, residence and region were significantly associated factors of RSB. Keeping all other individual and community level factors, urban dwellers were found to have about 2.8 times higher odds of RSB as compared to rural ones (AOR: 2.774, 95% CI: 1.789, 4.300). Individuals in Addis Ababa, Dire Dawa, Tigray, Amara, and Gambela regions were found to have higher odds of having RSB as compared to those who lives in Benshangul gumuz region of Ethiopia. These regions were also found to be partly hot spot areas of RSB as evidenced by the geo-statistical analysis (**Table 3**).

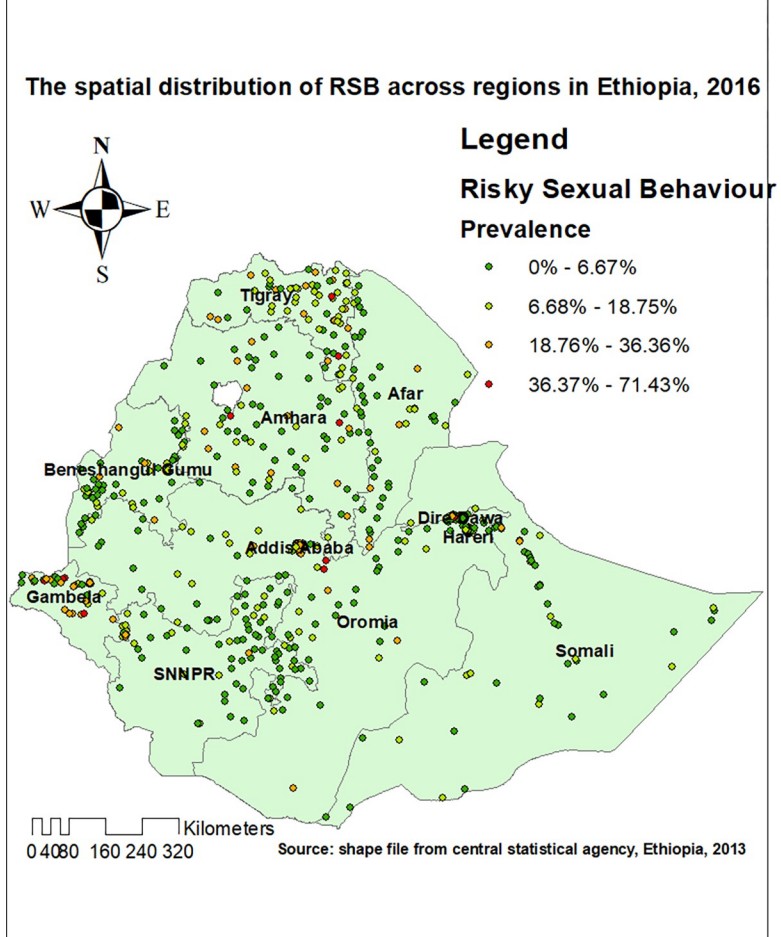

**Fig 3. Spatial distribution of risky sexual behviour in Ethiopia, 2016.**

## Discussion

Sexual behaviour is the most important aspect of human beings, for it can influence the health and productivity of the nation. Risky Sexual Behaviour is something which has to be investigated, so that a focused intervention can be made for better reproductive health of the community. Hence, this study has been aimed to assess the problem using both the geo-statistical and multilevel analysis, for the problem of interest is thought to be multi-dimensional in nature, affected by both the neighborhood effects and individual factors [32].

The finding of the study revealed that the prevalence of RSB was found to be 10.2% (95% CI: 9.64%, 10.81%). The magnitude ranged from 15.6 in some parts of Somalia, Benshangul and Addis Ababa to 48.5% in Tigray, Amhara and Oromia. Even though many policies and strategies have been designed and implemented in Ethiopia, RSB remains to be a great public health concern. The national level magnitude identified is lower than the findings of the studies conducted in Benshangul Gumuz, Harari regional state, Gondar, Mekelle, Axum, Bahir Dar, and systematic review and meta-analysis study conducted on college and university students [3,13,15,33–36]. This might be because of the difference in the size of the denominator, and study area. The social and cultural environment affects the human behavior either positively or negatively [2], a reason for regional variations in the magnitude of the RSB in

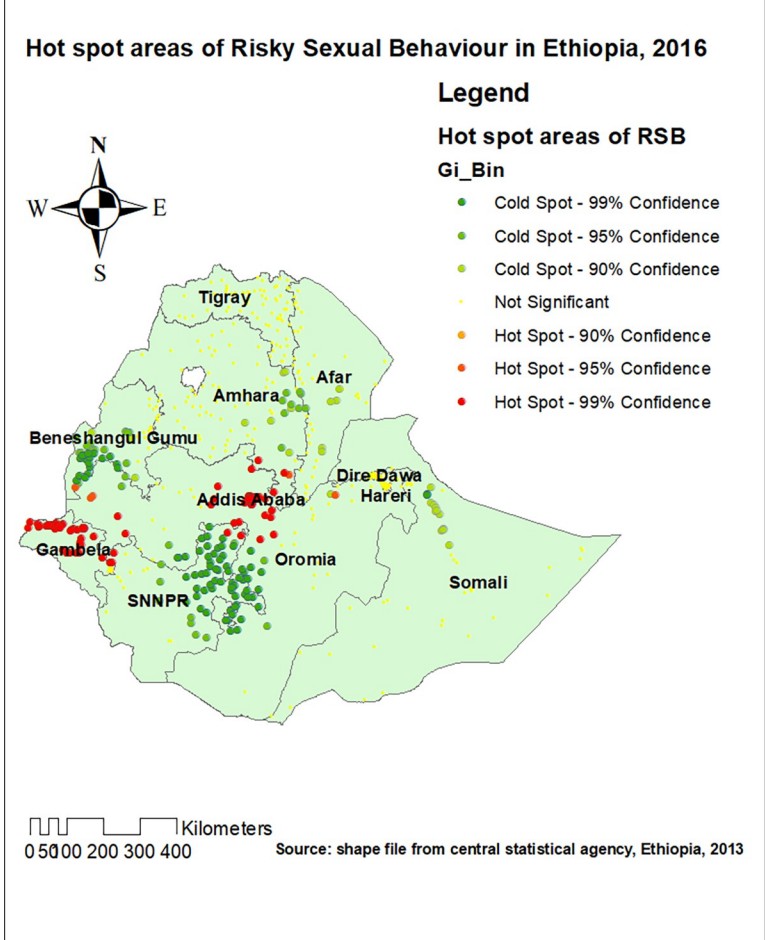

**Fig 4. The hot spot analysis of risky sexual behaviour in Ethiopia, 2016.**

Ethiopia. Besides, the magnitude identified is lower than the finding of the study conducted in sub-Saharan Africa, and Botswana [37,38]. However, the magnitude identified is still unacceptably high, and has to be given great infancies as far as creating healthy and productive generation is concerned in the country.

Furthermore, the spatial distribution of RSB in Ethiopia was found to be non-random. This was evidenced by the clustered spatial autocorrelation result of RSB (Moran's I statistics = 0.82, p-value = 0.001). The hot spot areas of were identified in Gambela, Addis Ababa, and western part of Dire Dawa. The identified significant spatial heterogeneity of RSB has pointed out the presence of disparities in terms of sexual and reproductive health policies and activities across the regions of Ethiopia. Besides, the variation in level of urbanization, change in the pattern of living and the increment of the population might be one of the reasons for the identified non-random spatial distribution of the RSB in Ethiopia.

In the spatial scan statistics analysis, 85 most likely clusters were detected spatially. The most likely clusters of RSB were detected in some parts of Addis Ababa, Gambela, and western part of Dire Dawa. From the most likely clusters, 53 of them were primary clusters located at 8.972907 N and 38.745529 E with 13.24 km radius. Individuals who live in the primary clusters were 3.26 times more likely to have RSB as compared to outside the window (RR = 3.26, LLR = 111.59, P-value < 0.001). Addis Ababa, the capital of Ethiopia, is the most crowed city

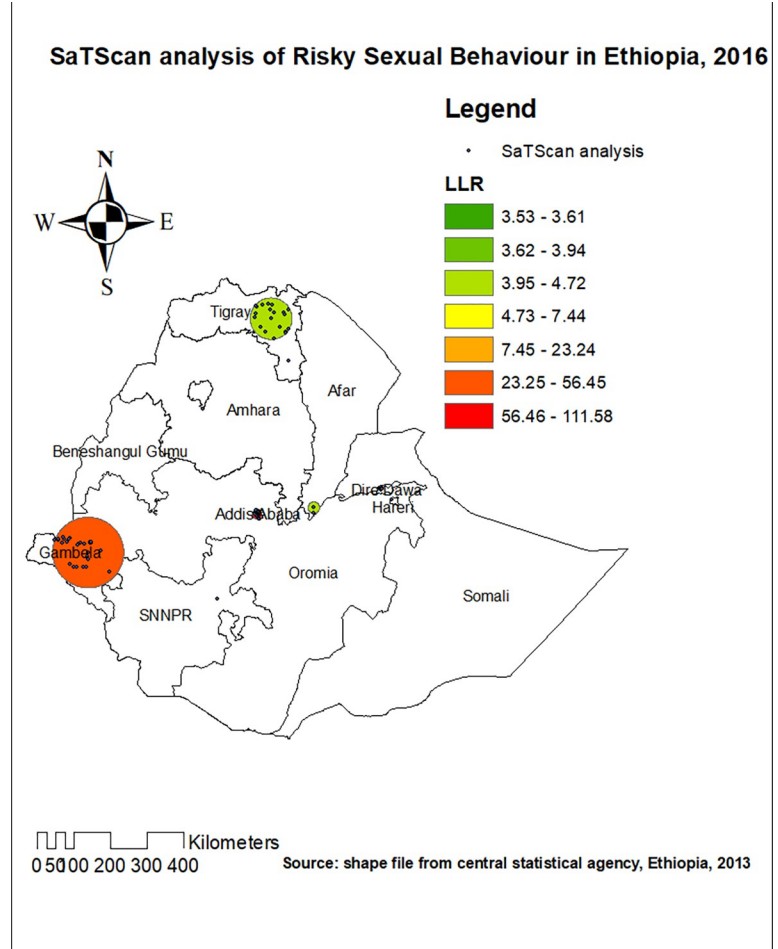

**Fig 5. The SaTScan analysis of significant most likely clusters of risky sexual behaviour in Ethiopia, 2016.**

of the country where individuals from different regions are living. Hence, RSB is expected in this city, for urbanization is one of the factors of increment in RSB. A more liberal way of living which is frequently seen in the identified hot spot areas could explain the identified high proportion of RSB.

In the mixed effect multivariable logistic regression analysis, significantly associated individual level factors identified were age, literacy level, smoking status, and ever heard of HIV/ AIDs. The finding has shown that as age increases the odds of RSB decreases by 2% (AOR: 0.978, 95% CI: (0.961 0.995)). Even though the difference is not too much, the finding has pointed out that adolescents should be given especial attention regarding to their sexual behaviour. These groups of individuals are vulnerable to many other social and behavioural activities which may affect their health conditions. RSB is the most important of all, for it could affect their reproductive health, fertility and productivity in their lives. This finding is also supported by the other study conducted South Africa and in Ethiopia on epidemiology of RSB among adolescents an [15,32]. Individuals with no formal education were found to have higher odds of RSB as compared to their counterparts. Understanding the consequence of unsafe sex is important not to engage in it. This requires literacy and the ability to capture information disseminated through different media. The finding is in line with the studies conducted Gahna,

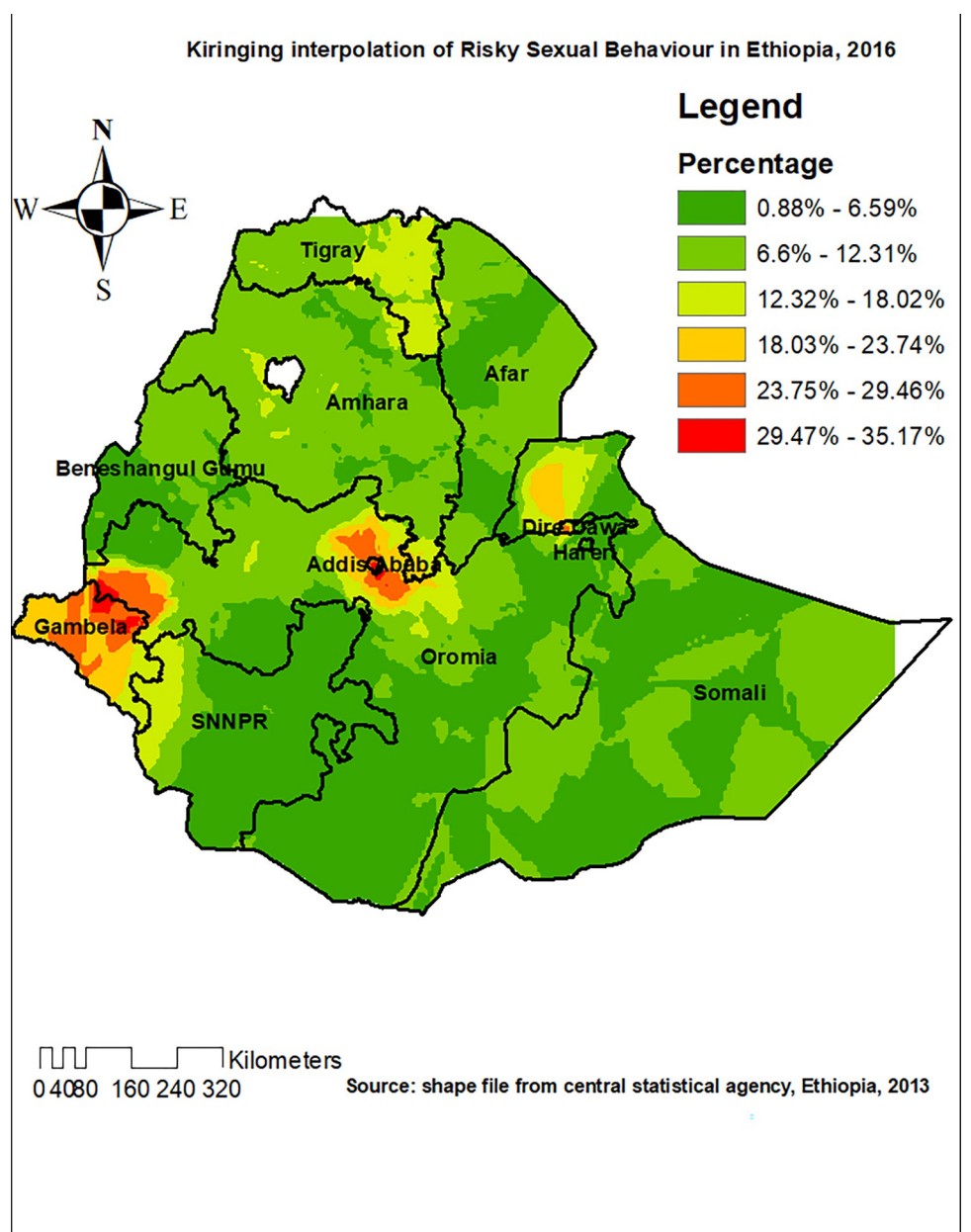

**Fig 6. The Kriging interpolation of risky sexual behaviour in Ethiopia, 2016.**

Ethiopia, and a meta-analysis study done in developing countries [39–41].The other significant individual level factor of RSB was smoking status. Smokers were found to have 3.3 times higher odds of having RSB as compared to non-smokers (AOR: 3.281; 95% CI: (1.247 8.631)). This shows that substance abuse or addiction is linked to RSB. A number of studies have also identified that alcohol and smoking are associated with increased number of sexual partners, early initiation of unsafe sex, and other risky sexual behaviours [4,42,43]. Besides, information about HIV/AIDS is important to prevent RSB among women and men in the age of 15–49 years, as evidenced by the finding of this study that the odds of having RSB among study subjects who have never heard of HIV/AIDS was 1.9 times higher as compared to their counterparts. Even though much has been done regarding to awareness creation regarding to HIV/

**Table 2. Random effect (community-level variability) and model fitness.**

| Parameters | Null model | Model 2 | Model 3 | Model 4 |
|---|---|---|---|---|
| **Measures of variation** | | | | |
| Community level variance (SE) | 4.50 | 3.82 | 2.69 | 2.85 |
| ICC | 0.253 | 0.135 | 0.094 | 0.077 |
| MOR | 2.736 | 1.978 | 1.744 | 1.649 |
| PCV | Reff | 15.1% | 67.3% | 57.9% |
| AIC | 47949.38 | 45583.81 | 47743.76 | 46417.2 |
| **Model comparison** | | | | |
| Deviance | 6405.0031 | 6087.4232 | 6105.9775 | 5902.5781 |
| LR test vs. logistic model | 336.06*** | 84.49*** | 55.29*** | 33.20*** |
| Log-likelihood ratio | -2728.11 | 2644.27 | -2639.62 | -2578.71 |
| LR test | ————— | M1 and M2 = **317.58***  | M2 and M3 = **298*** | M3 and M4 = **203.40*** |

\* = P-value < 0.05

\*\* = P value < 0.01

\*\*\* = P value < 0.001.

ICC Inter cluster correlation coefficient, MOR Median odds ratio, PCV proportional change in variance.

AIDS, still it is of greatest importance to keep disseminating information regarding to sexually transmitted infections like HIV/AIDS in Ethiopia.

Significantly associated community level factors were residence and region. Individuals who live in urban areas were found to have 2.8 times higher odds of RSB as compared to rural residents (AOR: 2.774; 95% CI: (1.789 4.300)). Obviously, there are risky social and behavioural events in urban areas. Cultural and norm diversity, peer pressures and other urbanization related factors affects the sexual behaviour of individuals. The expansion of night club industry and other recreational activities are thought to expose individuals in urban areas to unhealthy sexual activities. The influence of type of residence on the sexual behaviour of individuals is also reported by the study conducted in sub-Saharan Africa [44].

As supported by the geostatistical analysis part of this study, there is significant regional variation in terms of RSB among study subjects. Individuals in Addis Ababa, Dire Dawa, Gambela, Tigray, and Amhara regions of Ethiopia have 3.1, 1.7, 4.1, 3.3, and 2 times higher odds of RSB as compared to people who live in Benshangul gumuz region of Ethiopia, a cold spot region. Hence, re sensitization, and culture specific sexual health promotion activities has to be designed in the identified high-risk regions of Ethiopia, so that a more sexually healthy and productive community can be created.

Generally, this study has come up with new geo-statistical evidences on the distribution and factors of RSB in Ethiopia. It has strengths for it was done considering the hierarchical nature of the survey data, addressing the clustering effects and individual level factors of RSB. The two methodologies used (spatial and multilevel) enabled us to holistically understand the problem. However, the study was not without limitations, for it was done using a survey dataset in which it is impossible to establish a causal relationship between the factors and outcome of interest. Furthermore, since SaTScan analysis only detects circular and/or elliptical clusters, it's likely that clusters of other shapes go unreported.

## Conclusion and recommendation

This study showed that the spatial distribution of RSB was significantly varied across the country. The hotspot areas of RSB were located in the Addis Ababa, Dire Dawa, and Gambela

**Table 3. A multilevel analysis of factors associated with RSB among reproductive age women in Ethiopia, data from 2016 EDHS.**

| Variables | Model– 2 AOR, [95% CI] | Model– 3 AOR, [95% CI] | Model- 4 AOR, [95% CI] |
|---|---|---|---|
| Age (years) | 0.98 [0.964, 0.997] | | **0.978 [0.961 0.995]**[*] |
| Literacy level | | | |
| Literate | Ref | | Ref |
| No formal education | 0.623 [0.461 0.841] | | **1.31 (1.023 1.422)**[*] |
| Marital status | | | |
| Married | Ref | | Ref |
| Single | 1.549 [1.085 2.212] | | 1.308 [0.906 1.88] |
| Widowed/separated | 0.755 [0.433 1.315] | | 0.701 [0.416 1.182] |
| Wealth | | | |
| Poor | 1.084 [0.748 1.570] | | 1.092 [0.751 1.590] |
| Middle | Ref | | Ref |
| Rich | 1.211 [0.806, 1.819] | | 0.884 [0.561 1.395] |
| Age at first sex | | | |
| <18years | Ref | | Ref |
| > = 18 years | 1.512 [1.204 1.898] | | 1.488 [0.173 1.890] |
| Smoking status | | | |
| Not Smoker | Ref | | Ref |
| Smoker | 3.169 [1.180, 8.506] | | **3.281 [1.247 8.631]**[*] |
| Ever been tested for HIV | | | |
| Tested | Ref | | Ref |
| Not tested | 0.980 [0.712 1.350] | | 1.192 [0.849 1.676] |
| Media exposure | | | |
| Exposed | Ref | | Ref |
| Not exposed | 2.632 [0.685 4.111] | | 1.908 [0.804 1.024] |
| Heard of AIDS | | | |
| Yes | Ref | | Ref |
| No | 0.980 [0.712 1.350] | | **1.864 [1.088 3.195]**[*] |
| Community literacy | | | |
| Low | | Ref | Ref |
| High | | 1.340 [0.953 1.886] | 1.344 [0.914 1.977] |
| Residence | | | |
| Rural | | Ref | Ref |
| Urban | | 2.954 [2.160 4.039] | **2.774 [1.789 4.300]**[***] |
| Region | | | |
| Benshangul | | Ref | Ref |
| Tigray | | 2.681 [1.781 4.035] | **3.342 [2.153 5.191]**[***] |
| Afar | | 1.473 [0.872 2.486] | 1.641 [0.956 2.811] |
| Amhara | | 1.529 [0.967 2.418] | **1.981 [1.215 3.231]**[**] |
| Oromia | | 1.289 [0.806 2.062] | 1.364 [0.829 2.244] |
| Somali | | 1.018 [0.623 1.663] | 0.942 [0.553 1.607] |
| SNNPR | | 0.646 [0.384 1.087] | 0.703 [0.409 1.206] |
| Gambela | | 4.026 [2.531 6.404] | **4.100 [2.578 6.521]**[***] |
| Harari | | 0.842 [0.503 1.412] | 0.906 [0.535 1.531] |
| Addis Ababa | | 3.234 [1.986 5.270] | **3.0719 [1.830 5.156]**[***] |
| Dire Dawa | | 1.646 [0.990 2.738] | **1.696 [1.018 2.825]**[*] |

* = P-value < 0.05

** = P value < 0.01

*** = P value < 0.001.

region. Furthermore, age, literacy level, smoking status, ever heard about HIV/AIDS, residence and region were significantly associated with RSB. Hence, policymakers and other concerned bodies should design effective public health interventions to reduce the unacceptably high

magnitude of RSB and the associated morbidity and mortality. The problem shouldn't be under looked for it has the ability to touch each and every aspect of the lives of individuals and the future of the country in general. Besides, future researchers had better explore the problem using a qualitative approaches so that the interventions could be more specific and evidence based.

## Supporting information

**S1 File. The minimal anonymized dataset.**
(CSV)

## Acknowledgments

We would like to thank the measure DHS program for providing the dataset.

## Author Contributions

**Conceptualization:** Denekew Tenaw Anley, Melkamu Aderajew Zemene, Asaye Alamneh Gebeyehu, Getachew Asmare Adella, Gizachew Ambaw Kassie, Misganaw Asmamaw Mengstie, Mohammed Abdu Seid, Endeshaw Chekol Abebe, Molalegn Mesele Gesese, Yenealem Solomon, Natnael Moges, Berihun Bantie, Sefineh Fenta Feleke, Tadesse Asmamaw Dejenie, Ermias Sisay Chanie, Wubet Alebachew Bayih, Natnael Amare Tesfa, Wubet Taklual, Dessalegn Tesfa, Rahel Mulatie Anteneh, Anteneh Mengist Dessie.

**Data curation:** Denekew Tenaw Anley, Melkamu Aderajew Zemene, Asaye Alamneh Gebeyehu, Getachew Asmare Adella, Misganaw Asmamaw Mengstie, Endeshaw Chekol Abebe, Yenealem Solomon, Natnael Amare Tesfa, Rahel Mulatie Anteneh, Anteneh Mengist Dessie.

**Formal analysis:** Denekew Tenaw Anley, Melkamu Aderajew Zemene, Asaye Alamneh Gebeyehu, Getachew Asmare Adella, Yenealem Solomon, Natnael Moges, Sefineh Fenta Feleke, Tadesse Asmamaw Dejenie, Ermias Sisay Chanie, Natnael Amare Tesfa, Wubet Taklual, Dessalegn Tesfa, Rahel Mulatie Anteneh, Anteneh Mengist Dessie.

**Investigation:** Denekew Tenaw Anley, Getachew Asmare Adella, Dessalegn Tesfa.

**Methodology:** Denekew Tenaw Anley, Asaye Alamneh Gebeyehu, Natnael Atnafu Gebeyehu, Getachew Asmare Adella, Misganaw Asmamaw Mengstie, Mohammed Abdu Seid, Endeshaw Chekol Abebe, Molalegn Mesele Gesese, Yenealem Solomon, Natnael Moges, Berihun Bantie, Sefineh Fenta Feleke, Tadesse Asmamaw Dejenie, Wubet Alebachew Bayih, Natnael Amare Tesfa, Wubet Taklual, Dessalegn Tesfa, Anteneh Mengist Dessie.

**Software:** Denekew Tenaw Anley, Melkamu Aderajew Zemene, Natnael Atnafu Gebeyehu, Gizachew Ambaw Kassie, Misganaw Asmamaw Mengstie, Mohammed Abdu Seid, Endeshaw Chekol Abebe, Natnael Moges, Berihun Bantie, Dessalegn Tesfa, Rahel Mulatie Anteneh, Anteneh Mengist Dessie.

**Supervision:** Misganaw Asmamaw Mengstie.

**Validation:** Denekew Tenaw Anley, Gizachew Ambaw Kassie, Misganaw Asmamaw Mengstie, Molalegn Mesele Gesese, Tadesse Asmamaw Dejenie, Wubet Alebachew Bayih, Wubet Taklual, Rahel Mulatie Anteneh, Anteneh Mengist Dessie.

**Visualization:** Natnael Atnafu Gebeyehu, Endeshaw Chekol Abebe, Molalegn Mesele Gesese, Sefineh Fenta Feleke, Ermias Sisay Chanie, Wubet Alebachew Bayih, Wubet Taklual.

**Writing – original draft:** Denekew Tenaw Anley, Natnael Atnafu Gebeyehu, Gizachew Ambaw Kassie, Mohammed Abdu Seid, Berihun Bantie, Ermias Sisay Chanie.

**Writing – review & editing:** Denekew Tenaw Anley, Natnael Atnafu Gebeyehu, Gizachew Ambaw Kassie, Mohammed Abdu Seid, Berihun Bantie, Ermias Sisay Chanie.

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
