## [Decision Letter · Decision Letter 0]

13 Mar 2024

PONE-D-23-21383Hotspot areas of risky sexual behaviour and associated factors in Ethiopia: Further spatial and mixed effect analysis of Ethiopian Demographic Health SurveyPLOS ONE

Dear Dr. Anley,

Thank you for submitting your manuscript to PLOS ONE. After careful consideration, we feel that it has merit but does not fully meet PLOS ONE’s publication criteria as it currently stands. Therefore, we invite you to submit a revised version of the manuscript that addresses the points raised during the review process.

We look forward to receiving your revised manuscript.

Kind regards,

Mesfin Gebrehiwot Damtew (PhD)

Academic Editor

PLOS ONE

Journal Requirements:

4. Thank you for uploading your study's underlying data set. Unfortunately, the repository you have noted in your Data Availability statement does not qualify as an acceptable data repository according to PLOS's standards.

Reviewers' comments:

Reviewer's Responses to Questions

**Comments to the Author**

1. Is the manuscript technically sound, and do the data support the conclusions?

Reviewer #1: Yes

2. Has the statistical analysis been performed appropriately and rigorously? 

Reviewer #1: Yes

3. Have the authors made all data underlying the findings in their manuscript fully available?

Reviewer #1: Yes

4. Is the manuscript presented in an intelligible fashion and written in standard English?

Reviewer #1: Yes

5. Review Comments to the Author

Reviewer #1: Comments to author(s)

Manuscript Title: Hotspot areas of risky sexual behavior and associated factors in Ethiopia: Further spatial and mixed effect analysis of Ethiopian Demographic Health Survey.

Manuscript Number: PONE-D-23-21383

General comments:

The manuscript is well written, and the abstract, introduction, and other sections are clearly stated. The discussion and conclusions are in line with the key results. However, there are minor comments to be corrected, as suggested specifically below. The source of data for this study was 2016 DHS data, which is outdated, so there may have been many changes since then, and the findings might not show the recent events. The page numbers are not given.

Abstract:

1. Methods: On line 62, add the type of study design

Introduction:

2. On line 118, the authors did not include citations for previous studies conducted in Ethiopia. Please include a citation for existing evidence.

3. On lines 122-130, the authors did not elaborate well about the importance of exploring spatial analysis. What are the advantages of spatial analysis over other studies? Clearly describe the gaps and try to illustrate how the spatial analysis fulfills those gaps.

Methods and Materials:

4. On line 149, the study population is not clearly described.

5. The authors did not include the sub-sections in this manuscript. Sub-sections of eligibility criteria and data collection tools and procedures were missed. Please add these subsections in the methods part of the appropriate place.

6. The source of the data for this study was not also mentioned in the manuscript. Please describe the source of the data and put it in the proper place.

7. On line 152, what is the importance of sample weighting for DHS data analysis?

8. On lines 167 and 168, the authors did not put the operational definitions for community-level literacy and community-level poverty. However, you did not get these variables from the DHS data set; rather, they are composite variables developed from individual-level factors such as literacy status and wealth index. The problem of using these composite variables as level two without any evidence is multicollinearity. Repeated or twice-use of the same variables is unscientific. Therefore, I would like to recommend using only place of residence and region in the level two lists of variables.

9. On line 173, before spatial autocorrelation, please write spatial analysis as a sub-heading in the analysis. Then you can conduct step-by-step analysis: autocorrelation, hotspot analysis, interpolation, etc.

10. On line 210, why did you prefer to use LLR rather than deviance or AIC for model comparison? Which one is best for fitting multilevel binary logistic regression?

11. Data analysis: on line 234, what is the importance of fitting spatial regression using OLS in addition to fitting multilevel binary logistic regression? You can do spatial analysis with multilevel analysis or spatial analysis with spatial regression. Please choose either of the two.

12. After performing the OLS regression, the authors have reported that using GWR has poor model performance compared with the OLS output by using AIC and adjusted R square. The important question here is: Have you checked the important assumptions during the OLS regression analysis before comparing the model performance? The authors did not report the presence or absence of violations of important assumptions to choose either OLS regression or GWR. If you are interested in fitting spatial regression, please make assumptions. If the assumptions are violated, the model performance in the GWR is usually better than the OLS model. One important thing here is to not forget that this section is the methods part. Please don’t mix the result statements in the methods part. We cannot talk about model performance before data analysis; please rewrite this part as well. On the other hand, if you are interested in reporting spatial regression, please revise your title and objective in order to incorporate this.

13. On lines 243 and 244, this is one of the methods section parts, but the authors have written it as a result section. In the result section, you have reported that RSB was clustered across EAs with Moran’s index close to one (Moran’s I value of 0.82 (P-value <0.001). Moreover, your ICC report was 25%, which showed spatial heterogeneity between clusters. However, in the spatial regression analysis, you reported that there was no spatial heterogeneity of explanatory variables in RSB. Therefore, I have doubts that the spatial regression analysis was not conducted properly. I expect the presence of spatial non-stationarity. Therefore, I would like to recommend you re-analyze the spatial regression or report only spatial and multilevel analysis. However, the authors did not report OLS and GWR regression in the result section.

Results:

14. On lines 261 and 262, put the frequency and percentage figure with your text description.

15. On line 274, before sub-section' spatial autocorrelation', write first spatial analysis.

16. On lines 286-287: In the hotspot analysis output, why did you not report significant cold spots?

17. In table 2, what parameter did you use for model comparison, and which model was selected by what criteria?

18. According to your model analysis plan in the methods section, you did not perform the spatial regression analysis using OLS and GWR and you did not report the results the results section. Therefore, it is better to omit the paragraph statements in the methods section

Discussion:

19. On lines 372 to 381, please describe findings rather than directly results.

20. On lines 283-284, 397, and 409, please omit the AOR and 95% CI. The effect size is just enough.

21. At the end of the discussion, please add the strengths and limitations of your study.

Conclusion

22. On line 432, please replace the heading of the conclusion with “Conclusion and recommendation.

6. PLOS authors have the option to publish the peer review history of their article (what does this mean?). If published, this will include your full peer review and any attached files.

Reviewer #1: No

---

## [Author Response · Author response to Decision Letter 0]

3 Apr 2024

A rebuttal letter

Manuscript Title: Hotspot areas of risky sexual behavior and associated factors in Ethiopia: Further spatial and mixed effect analysis of Ethiopian Demographic Health Survey.

Manuscript Number: PONE-D-23-21383

Editor’s comments Authors’ response

Journal Requirements:

Thank you dear editor for your valuable comment. The revised manuscript is revised for the PLOS ONE’s style requirements.

 If you’ve not already done so, consider depositing your raw data in a repository to ensure your work is read, appreciated and cited by the largest possible audience. You’ll also earn an Accessible Data icon on your published paper if you deposit your data in any participating repository (https://plos.org/open-science/open-data/#accessible-data).

 Thank you dear editor for your comment. As per your comment, the data has been deposited in https://doi.org/10.5061/dryad.s1rn8pkgj

Thank you so much dear editor for your valuable comment. As per your comment, ethics statement is included in the ‘Methods’ section of the manuscript. What is written looks like “The data used for the analysis and the authorized letter were obtained from the measure DHS program website after proper registration and request. The data was accessed on Jan 30, 2023. Besides, authors had no access to information that could identify individual participants during or after data collection.” (page 9, line number 246 -249).

Besides the authorized letter is uploaded as supporting information for editor only file. 

4. Thank you for uploading your study's underlying data set. Unfortunately, the repository you have noted in your Data Availability statement does not qualify as an acceptable data repository according to PLOS's standards. At this time, please upload the minimal data set necessary to replicate your study's findings to a stable, public repository (such as figshare or Dryad) and provide us with the relevant URLs, DOIs, or accession numbers that may be used to access these data. For a list of recommended repositories and additional information on PLOS standards for data deposition, please see https://journals.plos.org/plosone/s/recommended-repositories.

 Thank you for the comments. As per you’ your comment, the minimal dataset used in this study has been deposited in the repository (https://doi.org/10.5061/dryad.s1rn8pkgj).

Comments from Reviewer #1

General comments: 

The manuscript is well written, and the abstract, introduction, and other sections are clearly stated. The discussion and conclusions are in line with the key results. However, there are minor comments to be corrected, as suggested specifically below. The source of data for this study was 2016 DHS data, which is outdated, so there may have been many changes since then, and the findings might not show the recent events. The page numbers are not given. Thank you dear reviewer for your encouraging and constructive comments. 

The reason why we have used the EDHS 2016 dataset is because there is no other recent full dataset. The EDHS 2019 dataset is mini dataset which doesn’t represent the targets of this study. Besides, behavioral change is very gradual and there may not be any significant change in risky sexual behavior and hence, we are assured that the information could reflect what actually exists currently. 

As per your comment, the revised manuscript is given page numbers.

 Abstract: 

1. Methods: On line 62, add the type of study design 

 Thank you dear reviewer for your comment. The revised manuscript has been edited for this and indicated by track changes. 

Introduction: 

2. On line 118, the authors did not include citations for previous studies conducted in Ethiopia. Please include a citation for existing evidence. Thank you dear reviewer for your comment. As per your comment, the previous studies have been cited in the revised manuscript. 

3. On lines 122-130, the authors did not elaborate well about the importance of exploring spatial analysis. What are the advantages of spatial analysis over other studies? Clearly describe the gaps and try to illustrate how the spatial analysis fulfills those gaps. Thank you dear reviewer for your comment. As per your suggestion, the need of spatial analysis particularly in risky sexual behavior is more elaborated in the revised manuscript and indicated by track changes. 

Methods and Materials: 

4. On line 149, the study population is not clearly described. 

 Thank you for your comments. The source population was all women in the age of 15-49 years in Ethiopia, and the study population was those who had sexual contact 12 months preceding the survey. 

5. The authors did not include the sub-sections in this manuscript. Sub-sections of eligibility criteria and data collection tools and procedures were missed. Please add these subsections in the methods part of the appropriate place.

 Thank you dear reviewer for your constructive comments. As per your comment, the necessary corrections have been made in the revised manuscript and indicated by track changes. 

6. The source of the data for this study was not also mentioned in the manuscript. Please describe the source of the data and put it in the proper place.

 Thank you dear reviewer for your constructive comments. “The data used for the analysis and the authorized letter were obtained from the measure DHS program website (www.dhsprogram.com) after proper registration and request”. This has been written in the revised manuscript and indicated by track changes. 

7. On line 152, what is the importance of sample weighting for DHS data analysis?

 Thank you dear reviewer for your crucial comment. Weighting in the DHS dataset is crucial because it adjusts for differences in the probability of selection and interview, ensuring that the sample is representative of the target population. By applying weights, under-sampled subpopulations are weighted up, while over-sampled subpopulations are weighted down, resulting in a more accurate representation of the population. Besides, it is important to perform more precise data analysis, as it compensates for potential biases introduced during sampling. This leads to more reliable and generalizable findings.

8. On lines 167 and 168, the authors did not put the operational definitions for community-level literacy and community-level poverty. However, you did not get these variables from the DHS data set; rather, they are composite variables developed from individual-level factors such as literacy status and wealth index. The problem of using these composite variables as level two without any evidence is multicollinearity. Repeated or twice-use of the same variables is unscientific. Therefore, I would like to recommend using only place of residence and region in the level two lists of variables.

 Thank you for your constructive comment. As you have mentioned community level literacy and community level poverty are composite variables. We used these variables to see the community level variation in terms of outcome of interest (RSB). Both of the variables were included in the descriptive analysis (Table 1). However, only community level literacy was included in the in the inferential statistics part where community level poverty was removed because of multi-collinearity. Hence, we need to assure that the issue raised wouldn’t influence our study result. 

9. On line 173, before spatial autocorrelation, please write spatial analysis as a sub-heading in the analysis. Then you can conduct step-by-step analysis: autocorrelation, hotspot analysis, interpolation, etc.

 Thank you dear reviewer for your effort to improve the quality of our manuscript. The necessary correction has been made in the revised manuscript. 

10. On line 210, why did you prefer to use LLR rather than deviance or AIC for model comparison? Which one is best for fitting multilevel binary logistic regression?

 Thank you for your valuable comment. For fitting multilevel binary logistic regression models, LLR is often preferred for comparing nested models, such as those with different levels of complexity in terms of random effects or fixed effects, which the case in our study. However, Deviance can be useful for comparing non-nested models or assessing goodness of fit. Besides, LLR is typically preferred for nested model comparison in likelihood-based modeling, including logistic regression, while AIC can be useful for more general model selection tasks.

11. Data analysis: on line 234, what is the importance of fitting spatial regression using OLS in addition to fitting multilevel binary logistic regression? You can do spatial analysis with multilevel analysis or spatial analysis with spatial regression. Please choose either of the two.

 Thank you for your insightful comments. As you have mentioned we did spatial analysis with multilevel analysis. The reason why we fitted OLS is to check eligibility for GWR. Because assumptions were not fulfilled, we didn’t go to GWR and hence, spatial with multilevel analysis was done. It is to indicate this that we have included the issue in the method part. However, we took your constructive comment and we have deleted the eligibility assessment statement from method section of the revised manuscript for it wouldn’t affect the clarity of the manuscript. 

12. After performing the OLS regression, the authors have reported that using GWR has poor model performance compared with the OLS output by using AIC and adjusted R square. The important question here is: Have you checked the important assumptions during the OLS regression analysis before comparing the model performance? The authors did not report the presence or absence of violations of important assumptions to choose either OLS regression or GWR. If you are interested in fitting spatial regression, please make assumptions. If the assumptions are violated, the model performance in the GWR is usually better than the OLS model. One important thing here is to not forget that this section is the methods part. Please don’t mix the result statements in the methods part. We cannot talk about model performance before data analysis; please rewrite this part as well. On the other hand, if you are interested in reporting spatial regression, please revise your title and objective in order to incorporate this. 

 Thank you for your comment. What we have answered in comment #11 can answer this comment too. What we did is simply checking if our data is eligible for Spatial regression (GWR). In doing so, the assumptions were not found to be fulfilled. As per your comment, we have removed the eligibility assessment from the method part, for this stud was spatial with multilevel analysis. 

13. On lines 243 and 244, this is one of the methods section parts, but the authors have written it as a result section. In the result section, you have reported that RSB was clustered across EAs with Moran’s index close to one (Moran’s I value of 0.82 (P-value <0.001). Moreover, your ICC report was 25%, which showed spatial heterogeneity between clusters. However, in the spatial regression analysis, you reported that there was no spatial heterogeneity of explanatory variables in RSB. Therefore, I have doubts that the spatial regression analysis was not conducted properly. I expect the presence of spatial non-stationarity. Therefore, I would like to recommend you re-analyze the spatial regression or report only spatial and multilevel analysis. However, the authors did not report OLS and GWR regression in the result section.

 Thank you dear reviewer for your constructive comments. What we did is simply checking if our data is eligible for Spatial regression (GWR). In doing so, the assumptions were not found to be fulfilled. As per your comment, we have removed the eligibility assessment from the method part, for this stud was spatial with multilevel analysis only. Besides, what is reported here in this study is the result of spatial and multilevel analysis. 

 Results: 

14. On lines 261 and 262, put the frequency and percentage figure with your text description. Thank you dear reviewer for your comment. As per your comment, the necessary correction has been made in the revised manuscript. 

15. On line 274, before sub-section' spatial autocorrelation', write first spatial analysis. Thank you dear reviewer for your comment. As per your comment, the necessary correction has been made in the revised manuscript. The changes made have been indicated by track changes. 

16. On lines 286-287: In the hotspot analysis output, why did you not report significant cold spots? Thank you dear reviewer for your comment. The cold spots have been also reported in the figure 4.

17. In table 2, what parameter did you use for model comparison, and which model was selected by what criteria? Thank you for your constructive comments. Deviance and LLR test were performed and the model with the lowest deviance and LR test was considered as the best fitting model as explained in page #14 of line 309 and 317-318. Hence, as it is indicated in Table 2, the model 4 (full model) which is the mixed effect model was found to be the best fitting model. 

18. According to your model analysis plan in the methods section, you did not perform the spatial regression analysis using OLS and GWR and you did not report the results the results section. Therefore, it is better to omit the paragraph statements in the methods section Thank you dear reviewer. As per your constructive comment, we have removed the eligibility assessment statement from method section of the revised manuscript for it wouldn’t affect the clarity of the manuscript.

Discussion: .

19. On lines 372 to 381, please describe findings rather than directly results. Thank you dear reviewer. The necessary modification has been made in the revised manuscript. 

20. On lines 283-284, 397, and 409, please omit the AOR and 95% CI. The effect size is just enough. Thank you dear reviewer. We preferred to report the 95% CI in addition to the effect size, as it enhances the clarity and precision of the estimates. 

21. At the end of the discussion, please add the strengths and limitations of your study. Thank you dear reviewer. The strength and limitation of the study is written in the revised manuscript. 

Conclusion 

22. On line 432, please replace the heading of the conclusion with “Conclusion and recommendation.

 Thank you dear reviewer. The necessary correction has been made in the revised manuscript and indicated by track changes.

---

## [Decision Letter · Decision Letter 1]

29 Apr 2024

Hotspot areas of risky sexual behaviour and associated factors in Ethiopia: Further spatial and mixed effect analysis of Ethiopian Demographic Health Survey

PONE-D-23-21383R1

Dear Dr. Anley,

We’re pleased to inform you that your manuscript has been judged scientifically suitable for publication and will be formally accepted for publication once it meets all outstanding technical requirements.

Kind regards,

Mesfin Gebrehiwot Damtew (PhD)

Academic Editor

PLOS ONE

Additional Editor Comments (optional):

Reviewers' comments:

Reviewer's Responses to Questions

**Comments to the Author**

1. If the authors have adequately addressed your comments raised in a previous round of review and you feel that this manuscript is now acceptable for publication, you may indicate that here to bypass the “Comments to the Author” section, enter your conflict of interest statement in the “Confidential to Editor” section, and submit your "Accept" recommendation.

Reviewer #1: All comments have been addressed

2. Is the manuscript technically sound, and do the data support the conclusions?

Reviewer #1: Yes

3. Has the statistical analysis been performed appropriately and rigorously? 

Reviewer #1: Yes

4. Have the authors made all data underlying the findings in their manuscript fully available?

Reviewer #1: Yes

5. Is the manuscript presented in an intelligible fashion and written in standard English?

Reviewer #1: Yes

6. Review Comments to the Author

Reviewer #1: The authors have almost addressed the comments and questions. I have no additional comments here. I would like to say congratulations to the authors. I will endorse this manuscript for publication.

7. PLOS authors have the option to publish the peer review history of their article (what does this mean?). If published, this will include your full peer review and any attached files.

Reviewer #1: No

---

## [Editor Report · Acceptance letter]

22 May 2024

PONE-D-23-21383R1 

PLOS ONE

Dear Dr. Anley, 

I'm pleased to inform you that your manuscript has been deemed suitable for publication in PLOS ONE. Congratulations! Your manuscript is now being handed over to our production team.

Kind regards, 

on behalf of

Dr. Mesfin Gebrehiwot Damtew 

Academic Editor

PLOS ONE